# How Do Morphological Factors Influence the Green Nut Yield of Chinese Torreya?

**Xi Chen, Shangbin Bai and Dongming Fang \***

Jiyang College of Zhejiang A&F University, Zhuji 311800, China
\* Correspondence: dmfang@zafu.edu.cn

**Abstract:** As an important economic tree species, Chinese Torreya (*Torreya grandis* cv Merrillii) has been widely planted in the subtropical regions of China. However, it remains to be studied whether morphological traits are the key factors reflecting or affecting the green nut yield of Chinese Torreya, which is necessary for breeding research and plantation management. Therefore, in Zhuji in the Zhejiang Province, the central production area of Chinese Torreya, we investigated the morphological traits (height, ground diameter, under-crown height, crown width, and branching amount) and green nut yield of 120 randomly selected Chinese Torreya. Our results indicated that the differences in the morphological traits among Chinese Torreya individuals were relatively small, but those in the green nut yield traits were great. There was highly significant ($p < 0.01$) correlation between green nut yield and crown area and between green nut yield and root collar diameter (ground diameter). A moderate relationship ($r = 0.38$; $p < 0.05$) was observed between green nut yield and crown area, while a weak relationship ($r = 0.294$; $p < 0.05$) was detected between green nut yield and ground diameter. Tree height and branching amount had positive effects on green nut yield through other morphological traits, and under-crown height had indirect negative effects on green nut yield. Linear regression analysis showed a significant linear positive correlation between green nut yield and crown area, ground diameter, and crown width in the north–south and east–west directions ($p < 0.01$). These findings imply that if the tree height is fixed, increasing the ground diameter and crown area, appropriately increasing the branching amount, and reducing the under-crown height could be potential technical measures to improve the green nut yield of Chinese Torreya. Our study provides background information on green nut yield and its morphological traits in Chinese Torreya.

**Keywords:** correlation; nut production; morphological traits; *Torreya grandis* cv Merrillii

## 1. Background

Due to the multiple utilities (e.g., food, oil, and medicine) of its nutritious nuts [1], Chinese Torreya (*Torreya grandis* cv Merrillii) has been cultivated for more than 1300 years and has been vigorously promoted as a valuable economic tree species in the last 30 years by local governments in southern China, especially in Zhejiang province [2,3]. In the central production area of Chinese Torreya, such as Zhuji county of Zhejiang province, the Chinese Torreya industry has an annual production value of hundreds of millions of renminbi, accounting for nearly 80% of the local total agricultural income [4]. However, it is difficult for many farmers to master Chinese Torreya's growth habits and cultivation techniques, leading to a long-term low green nut yield without economic income. Therefore, improving the quality and yield of Chinese Torreya nuts has always been the central concern of local farmers and one of the most important research topics for relevant researchers [5]. For an improvement in green nut yield, it is necessary to know which kind of factors influence green nut yield.

For the green nut yield of Chinese Torreya, researchers have investigated the effects of harvest time [6], fertilization coupled with nitrogen deposition [7], fertilization with boron, zinc, copper, and molybdenum [8] or magnesium [9], biochar addition into the soil [10,11],

drought and shading [12], and meteorological factors [13]. In summary, existing research has mainly focused on the environmental impact and fertilizer application. In contrast, there are relatively few studies relating the morphological factors with the green nut yield of Chinese Torreya, despite morphological factors having been found to have positive correlations with the green nut yield of many other species [14–17]. Thus far, we have found only one study reporting the green nut yield model of ancient Chinese Torreya [18]. Still, the correlation between the green nut yield and morphological characteristics of artificial planting Chinese Torreya has not been reported.

Previous studies on other species have shown that the green nut yield is closely related to its morphological traits [14–17]. However, there are still significant differences among species, even between subspecies of the same species. For example, moderate but positive correlations between tree size and green nut yield have been reported for *Sclerocarya birrea* (A.Rich.) Hochst. ssp. *caffra* (Sond.) Kokwaro [15,16]. In contrast, only a weak positive relationship was found in another subspecies (*S. birrea ssp. Birrea*) of the same species [14]. Therefore, species-specific relationships between green nut yield and morphological traits are necessary to measure for different species, which is an essential pre-study for seedling breeding and implementing production management technology.

The importance of Chinese Torreya's morphological traits for its green nut yield is not evident. Therefore, it remains to be studied whether morphological traits are the key factors reflecting or affecting the green nut yield of Chinese Torreya. We recognize the critical role of multi-year monitoring on the dynamics of green nut yield. Still, in this study, we attempted to conduct a one-year (2019) trial exploration and build a background reference to the further and more complicated survey. Therefore, in this study, we investigated only a one-year green nut yield of Chinese Torreya, but as many morphological traits as possible (e.g., tree height, ground diameter, under-crown height, and crown width). We aimed to clarify the correlation between green nut yield and morphological traits based on analyzing these data. We hope that this study's findings may provide a fundamental theoretical basis for improving the cultivation technology of Chinese Torreya, which may further benefit the prediction, evaluation, and selection of Chinese Torreya to some extent.

## 2. Materials and Methods

### 2.1. Overview of the Experimental Site

The investigation site is in Tangzhouling of Zhuji, Zhejiang province (120°16′ E, 29°73′ N), which belongs to the subtropical monsoon climate zone with high temperatures and relatively low precipitation in summer [19]. The average temperature is 16.3 °C, the average yearly rainfall is 1373.6 mm, the relative humidity is 82%, the annual sunshine is 1887.6 h, and the average annual frost-free period is 233 days. The soil type is yellow–red soil with a thickness of 40~80 cm and a pH of 4.5~6.5. The site faces southeast, mainly with an altitude of 10~50 m and a slope of 10~30°. All Chinese Torreya planted were 3 + 2 (five-year) grafted seedlings transplanted in 2008. The plantation occupies an area of 10 ha, with a density of 600 trees per ha and 15 male trees per ha.

### 2.2. Measurement or Calculation of the Morphological Traits and Green Nut Yield

One-hundred-and-twenty Chinese Torreya trees were randomly selected at the experimental site in early September 2019. The tree height, ground diameter, under-crown height, and crown width of each sampled tree were measured. The crown area ($m^2$; Crown_area) was calculated with the measured width in the east–west and north–south directions (Crown_width_ew and Crown_width_ns; Equation (1)).

$$Crown\_area = \pi \times (Crown\_width\_ew/2) \times (Crown\_width\_ns/2) \tag{1}$$

The total fresh weight of green nuts per tree was weighed immediately after harvesting. Then, 100 green nuts were randomly selected from each tree and weighed (weight_per_100_nuts). Next, we measured the lengths and widths of 10 randomly sam-

pled green nuts from each tree. The volume of each green nut (Green_nut_volume ) was calculated with the measured length and width of each sampled green nut (Equation (2)).

$$Green\_nut\_volume = \pi \times Green\_nut\_length \times ((Crown\_width\_ew + Crown\_width\_ns)/2)^2 \quad (2)$$

*2.3. Data Processing and Analysis Methods*

Before examining the effects of morphological traits on green nut yield per tree and the weight$_{per\_100\_nuts}$, we first plotted them in the histogram to visually observe their distribution. Next, we tested if they were normally distributed with the Shapiro–Wilk normality test. If the green nut yield per tree was not normally distributed, the following statistical examinations (Spearman's test on correlations between traits and linear regressions between green nut yield and morphological traits) were conducted with nonparametric methods. Then, the correlation among traits was tested with a nonparametric test (Spearman's test), and a correlation coefficient matrix was plotted with the ggcorrplot package in R 4.2.0.

To avoid the disturbance of multicollinearity as much as possible, we applied two types of analyses to predict the green nut yield per tree with multiple morphological traits in SAS 9.4. One attempt was to examine multicollinearity and remove the variable with the most significant variance inflation value and lowest tolerance value until all independent variables had a variance inflation value of <10 and a tolerance value of >0.1. Then, the left traits were used to conduct a stepwise linear regression to predict the green nut yield per tree.

It was also attempted to conduct principal component regression, which is thought to be able to weaken the influence of multicollinearity. At first, principal component analysis was performed to produce several components combined with the analyzed traits. Then, the components with larger eigenvalues were used to predict the green nut yield per tree. At last, the fitting parameters of the components were converted to that of the original independent variables. The principal component regression was realized by the procedure PROC REG, specifying the option of PCOMIT in SAS 9.4.

As we found that the morphological traits were not able to predict the green nut yield fully, we further explored if the slope aspect and position contributed to the variance of green nut yield with the Kruskal–Wallis test in R 4.2.0.

## 3. Results and Analysis

*3.1. Individual Differences in the Morphological Traits and Green Nut Yield of Chinese Torreya*

The investigated Chinese Torreya had an average height of $3.4 \pm 0.6$ m, a ground diameter of $13.6 \pm 3.4$ cm, and a crown width of $3.3 \pm 0.8$ m$^2$. Their average under-crown height was $16.7 \pm 9.1$ cm and the average branching amount was $2.9 \pm 0.8$ (Table 1). The variation in morphological traits was as follows: tree height (17.8%) < ground diameter (25.2%) < crown area (25.8%) < branching amount (29.0%) < under-crown height (54.4%). Most of the morphological traits were normally distributed ($p > 0.05$), except for branching amounts and under-crown height ($p < 0.05$; Figure 1).

**Table 1.** Basic statistics for morphological and green nut yield traits of Chinese Torreya.

| Statistics | Tree Height/m | Ground Diameter/cm | Crown Area/m$^2$ | Under-Crown Height /cm | Branching Amount | Weight$_{per\_100\_nuts}$ /kg | Green Nut Volume/cm$^3$ | Yield per Tree/kg |
|---|---|---|---|---|---|---|---|---|
| Mean ± std | 3.4 ± 0.6 | 13.6 ± 3.4 | 3.3 ± 0.8 | 16.7 ± 9.1 | 2.9 ± 0.8 | 1.0 ± 0.1 | 12.3 ± 1.9 | 1.8 ± 1.6 |
| Range | 2.0–4.8 | 7.3–25.0 | 1.5–6.3 | 5–60 | 2–6 | 0.6–1.3 | 7.9–21.0 | 0.1–7.5 |
| CV (%) | 17.8 | 25.2 | 25.8 | 54.4 | 29.0 | 11.6 | 15.4 | 86.7 |

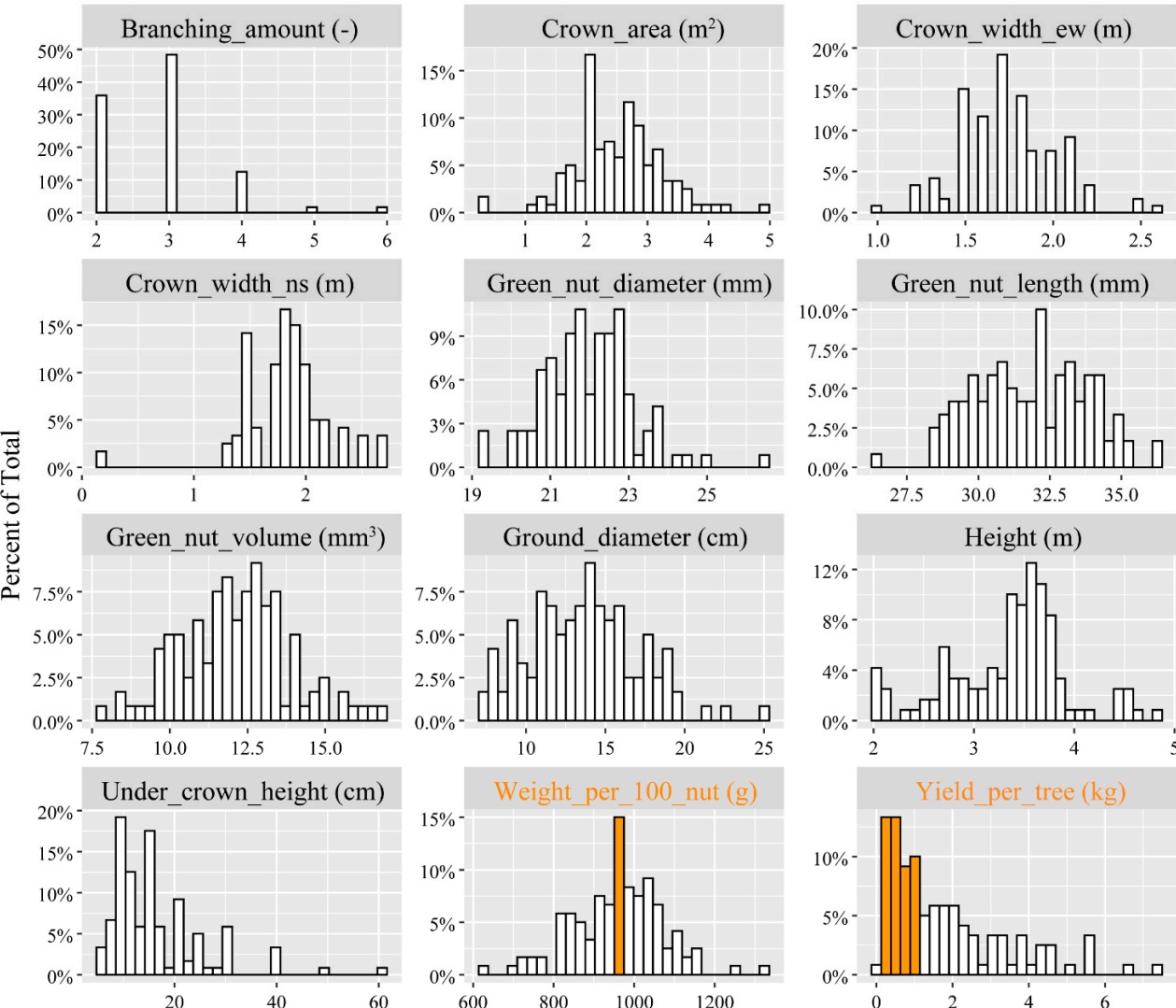

**Figure 1.** Distribution of traits. Note that the green nut yield per tree is not normally distributed (*p* < 0.05; checked with a Shapiro–Wilk test).

The average weight$_{per\_100\_nut}$ was approximately 1.0 ± 0.1 kg and the average green nut volume was 12.3 ± 1.9 cm$^3$. The green nut yield per tree was 1.8 ± 1.6 kg, but it lay within a wide range (0.1–7.5 kg) with a high variation (CV = 86.7%). The results suggest that the measured Chinese Torreya trees had a similar green nut size but varied in the whole-tree green nut yield. The weight$_{per\_100\_nuts}$ followed a normal distribution (*p* > 0.05), while the green nut yield per tree was skewed to the left (*p* < 0.05; Figure 1).

*3.2. Correlation Analysis among Morphological Traits*

As examined by the Shapiro–Wilk normality test, some traits were not normally distributed. Therefore, we explored the correlation between traits with a nonparametric test (Figure 2). We found that ground diameter was the only trait that correlated significantly with all the other traits, and only one of which (under-crown height) had a negative correlation with the ground diameter. Moreover, height had significant positive correlations with three crown traits (crown area and east–west and north–south crown widths; Figure 2).

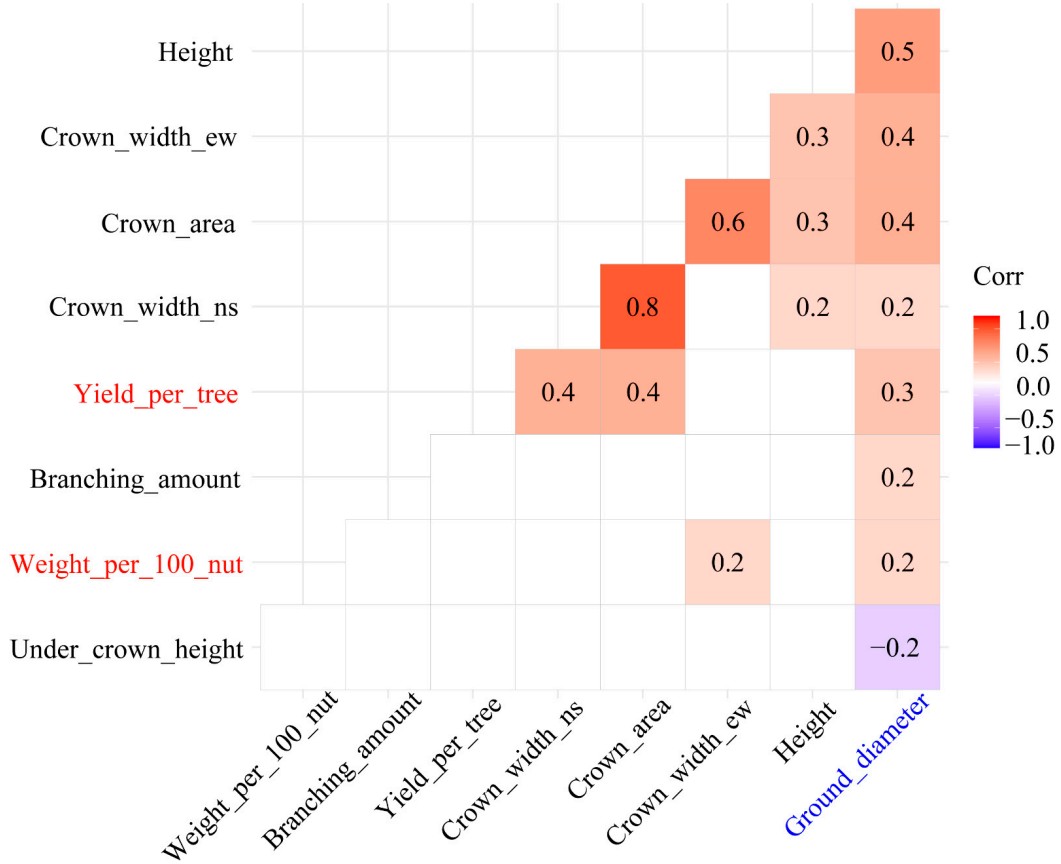

**Figure 2.** Correlation coefficients among traits were examined with a nonparametric test (Spearman test). The red and light purple colors in the grids indicate significant positive and negative correlations, respectively. The significance level is 0.05. The digit number in each cell is the correlation coefficient, which ranges from −1 to 1.

### 3.3. Relationship between Morphological Traits and Green Nut Yield

The Spearman's test indicated that green nut yield per tree was positively correlated with north–south crown width, crown area, and ground diameter (correlation coefficients were 0.4, 0.4, and 0.3, respectively; $p < 0.05$; Figure 2). In contrast, weight$_{per\_100\_nuts}$ was positively correlated with the east–west crown width and ground diameter (both correlation coefficients were 0.2; $p < 0.05$; Figure 2). A linear correlation test also confirmed the weak positive relationships between the green nut yield per tree and the three above morphological traits ($p < 0.05$; Figure 3) and the weak positive correlation between the east–west crown width and weight$_{per\_100\_nuts}$ ($p < 0.05$; Appendix A Figure A1).

Through both the stepwise (Table 2) and PCA (primary component analysis) linear multiple regression analyses (Table 3), we found that both the ground diameter and under-crown height had the most influential power on the green nut yield per tree ($R^2 = 0.29$; $p < 0.01$). The ground diameter and under-crown height explained 64.4% and 21.5% of the model variation, respectively (Table 2).

As we found that the morphological traits were not able to predict the green nut yield fully, we further explored if the slope aspect and position contributed to the variance of the green nut yield. However, neither the slope aspect nor position had an effect on the green nut yield per tree (Appendix A Figures A2 and A3).



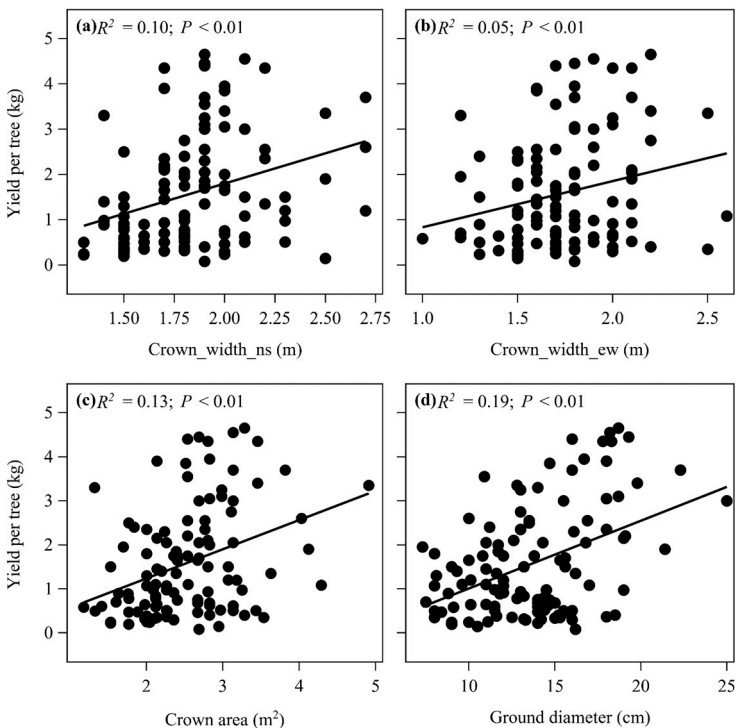

**Figure 3.** Relationship between green nut yield per tree and (**a**) north–south crown width (m), (**b**) east–west crown width (m), (**c**) crown area (m$^2$), and (**d**) ground diameter (cm).

**Table 2.** Stepwise linear multivariate model predicting green nut yield per tree with morphological factors.

| ProbF | Model R-Square | Variable Entered | Parameter Estimate | Partial R-Square | var_Explained |
|---|---|---|---|---|---|
| <0.01 | 0.29 | ground_diameter | 0.18 | 0.19 | 64.4% |
| | | under_crown_height | 0.03 | 0.06 | 21.5% |
| | | crown_width_NS | 0.83 | 0.03 | 8.61% |
| | | height | −0.30 | 0.02 | 5.47% |

Note: model R-square is the total R$^2$ of the multiple regression model; variable entered is the variable entered into the model; parameter estimate is the fitting parameter value of each variable; partial R-square is the partial correlation R$^2$ of each variable; var_ Explained is the proportion of each variable to the total R$^2$ of the model, representing the importance of the variable in the model.

**Table 3.** PCA (primary component analysis) linear multivariate model predicting green nut yield per tree with morphological factors.

| ProbF | Model R-Square | Variable | Estimate | Pr > \|t\| |
|---|---|---|---|---|
| <0.01 | 0.32 | ground_diameter | 0.19 | <0.01 |
| | | under_crown_height | 0.03 | 0.02 |
| | | height | −0.36 | 0.07 |
| | | branching_amount | −0.19 | 0.15 |
| | | crown_area | 1.94 | 0.20 |
| | | crown_width_EW | −2.78 | 0.21 |
| | | crown_width_NS | −1.76 | 0.40 |

Note: model R-square is the total R$^2$ of the multiple regression model; variable is the variable entered into the model; parameter estimate is the fitting parameter value of each variable; ProbF and Pr > \|t\| represent *p*-value of the model and each variable ($\alpha = 0.05$).

## 4. Discussion

The differences in the morphological traits among Chinese Torreya individuals were relatively small, and the green nut yield was significantly different among individuals, with the variation coefficient reaching 86.74%. The results indicate that the genetic diversity level of the green nut yield was higher than that of the morphological traits. The relatively large genetic coefficients of variation also indicated high expected genetic gains [20]. It is of great significance for the breeding and genetic improvement of Chinese Torreya cultivars aiming to reach the maximum green nut yield.

The results of the correlation analysis among the traits showed that there was a significant positive correlation between the ground diameter, tree height, and crown area, which is consistent with the research results of Xie et al. [18]. This suggests that as trees age, more of the nutrients absorbed are used in the reproductive process to facilitate the formation and development of fruit. The ground diameter and crown area were the main factors affecting the green nut yield. The tree height and branch amount had positive effects on the green nut yield through other morphological traits, while the under-crown height had indirect negative effects on the green nut yield. This is consistent with the theory of "internal correlation" believed by tree physiologist Paul [21]. That is, in addition to the influence of environmental factors on tree growth, there is also a relationship of mutual promotion and mutual restriction between the growth parts of trees.

In general, there may be some ecological or evolutionary balance between fruit size and green nut yield [22]. In this study, we found a very significant negative correlation between the green nut yield and green nut volume in Chinese Torreya, supporting the theoretical prediction that there is a trade-off between the green nut yield and fruit size [23]. In addition, we did not find a significant correlation between the fruit size and morphological traits of Chinese Torreya, which is consistent with our previous findings [1]. These results do not support the conclusion from other studies that tree body size is related to fruit size [24,25].

The linear regression analysis showed that the green nut yield of Chinese Torreya was significantly positively correlated with the ground diameter and crown area, as well as the crown width in the north–south and east–west directions, and the correlation coefficients ($R^2$) were all above 0.64. Through linear multiple regression analysis, we found that ground diameter has the greatest effect on Chinese Torreya's green nut yield. Strong and positive correlations are useful in tree breeding and selection.

## 5. Conclusions

Among all of the investigated morphological traits, the ground diameter was the most significant positive influencing factor of the green nut yield of Chinese Torreya. Additionally, the ground diameter significantly and positively correlated with all of the other traits except the under-crown height, which may indicate that other morphological traits might indirectly affect the yield via the ground diameter. Through linear multiple regression analysis, we found that the two factors of north–south crown width and ground diameter entered the final model of green nut yield prediction of Chinese Torreya ($p < 0.01$). However, the two factors only explain 23% of the green nut yield variation, which indicates that the green nut yield of Chinese Torreya may also be affected by other factors not monitored in this study. We speculate that this may be due to the differences in the microclimate and soil fertility of Chinese Torreya individuals, which need to be further studied in the future.

**Author Contributions:** Conceptualization, X.C.; methodology, X.C.; software, D.F.; validation, X.C., D.F. and S.B.; formal analysis, X.C. and D.F.; investigation, X.C.; resources, X.C.; data curation, X.C. and D.F.; writing—original draft preparation, X.C.; writing—review and editing, X.C., D.F. and S.B.; visualization, D.F.; supervision, X.C. and S.B.; project administration, X.C. and S.B.; funding acquisition, X.C. and D.F. All authors have read and agreed to the published version of the manuscript.

**Funding:** This research was supported by the Talent Project of Jiyang College of Zhejiang A&F University (grant numbers: 05251700035 and 05251700038).

**Data Availability Statement:** The data presented in this study are available on request from the corresponding author.

**Acknowledgments:** The authors are thankful to Ziyan Yu and Xin Chen for their help in field measurement and to Zhongqi Wu for allowing permissions.

**Conflicts of Interest:** The authors declare that they have no known competing financial interests or personal relationships that could have appeared to influence the work reported in this paper.

## Appendix A

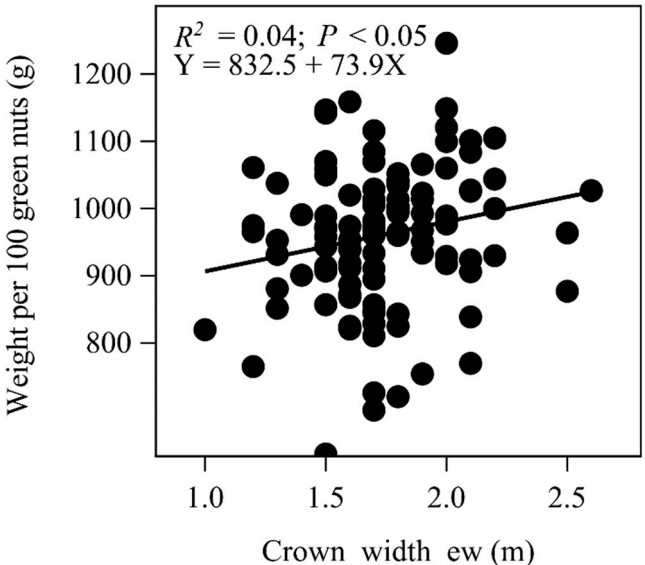

**Figure A1.** Relationship between weight per 100 green nuts and east–west crown width.

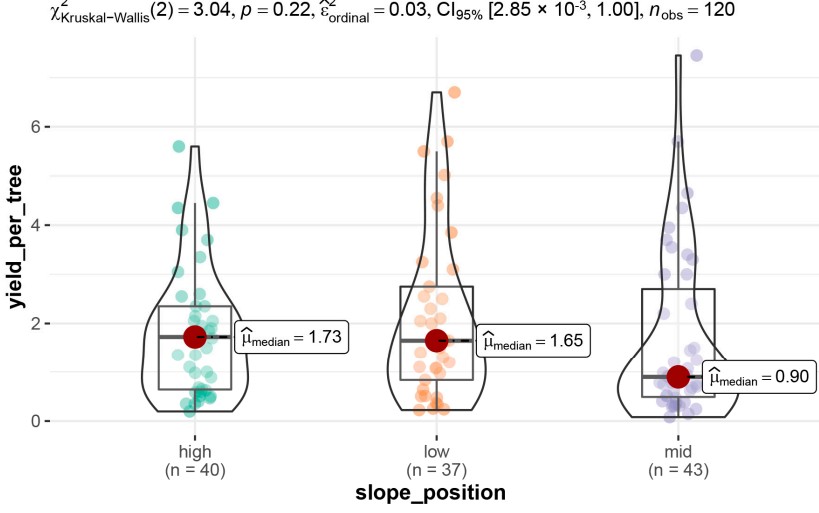

**Figure A2.** Comparison of green nut yield per tree (kg) among the different slope positions (high, low and mid). There is no significant difference among slope positions ($p > 0.05$).

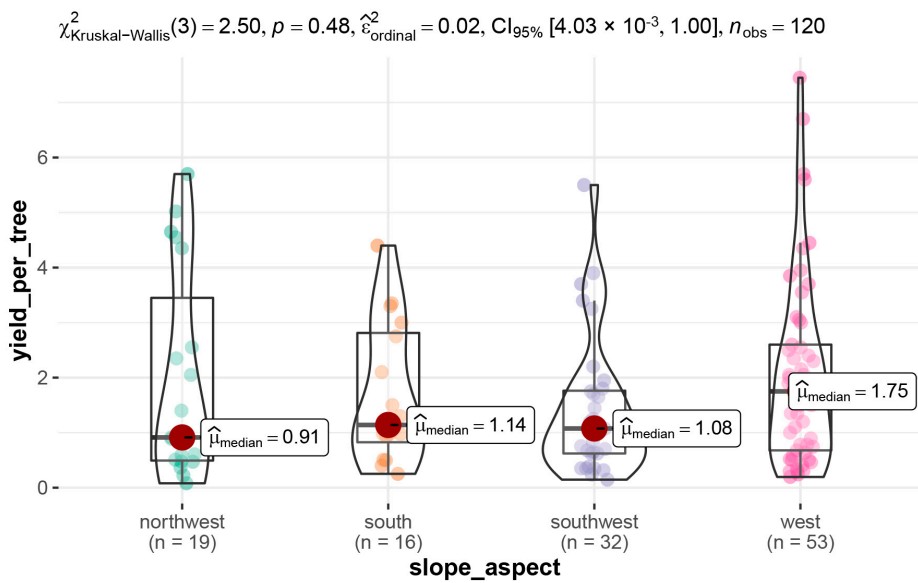

$\chi^2_{\text{Kruskal-Wallis}}(3) = 2.50,\ p = 0.48,\ \hat{\varepsilon}^2_{\text{ordinal}} = 0.02,\ \text{CI}_{95\%}\ [4.03 \times 10^{-3}, 1.00],\ n_{\text{obs}} = 120$

Pairwise test: **Dunn test**, Comparisons shown: **only significant**

**Figure A3.** Comparison of green nut yield per tree (kg) among the different slope aspects (northwest, south, southwest, west). There is no significant difference among slope aspects ($p > 0.05$).

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
