# Peer review of "How Do Morphological Factors Influence the Green Nut Yield of Chinese Torreya?"

_horticulturae, doi:10.3390/horticulturae9020202_

Round 1

Reviewer 1 Report

The article is interesting and generally well written and should be useful to researchers in the area. The technical work appears to be of high quality and the statistical treatment is robust. Some minor revisions to English are required. More specific comments follow.

1. There is some repetition in lines 11-12 cf. lines 14-15.

2. Line 14 and elsewhere - "yield" should be written as "nut yield" or "green nut yield" throughout the manuscript. Are yield and 100-nut weight expressed on an as-is moisture basis? Are moisture content values available?

3. Lines 18-19 - fruit yield is the same as nut yield? Why the change in terminology?

4. Lines 19-20 - Are the correlations significant? This should be indicated.

5. Line 96 - what was the methodology for determining yield of green nuts? 

6. Lines 102-103 - is there a difference between "yield" and "weight of green nuts"?

7. Line 133 - "mounts" should be changed to "amounts".

8. Lines 149-150, 151 - are these correlations significant? This should be indicated.

9. Figure 1 - is under crown height normally distributed or skewed?

10. Figure 2 - is there a significant negative correlation, i.e. is -0.2 significant and the light purple shade is to be interpreted as blue?

11. Figure 4 - the correlations are significant but very small in magnitude - are they of practical significance?

Author Response

Dear reviewer,

Thank you very much for the valuable comments and suggestions. We have revised the manuscript based on your suggestions as following. Moreover, the revised munuscript has been sent for an English check in MDPI.

  1. There is some repetition in lines 11-12 cf. lines 14-15.

Reply: We have deleted the sentence in new line 14: “Its central production area is in Zhuji of Zhejiang Province.”

  1. Line 14 and elsewhere - "yield" should be written as "nut yield" or "green nut yield" throughout the manuscript. Are yield and 100-nut weight expressed on an as-is moisture basis? Are moisture content values available?

Reply: we changed “yield” and “nut yield” to “green nut yield” throughout the manuscript. green nut yield and 100-nut weight are fresh weight on a same moisture basis. The fresh weight of green nuts per tree was weighed immediately after harvesting, and the moisture content was not measured.

  1. Lines 18-19 - fruit yield is the same as nut yield? Why the change in terminology?

Reply: we change “fruit yield” to “green nut yield”.

  1. Lines 19-20 - Are the correlations significant? This should be indicated.

Reply: the correlations were significant. We changed the sentences in new lines 22-23:

“A moderate relationship (r=0.38; P<0.05) was observed between green nut yield and crown area, while a weak relationship (r=0.294; P<0.05) was detected between green nut yield and ground diameter.”.

  1. Line 96 - what was the methodology for determining yield of green nuts?

Reply: We have redescribed in line 92-94 “The total fresh weight of green nuts per tree was weighed immediately after harvesting. Then 100 green nuts were randomly selected from each tree and weighed (Weight_100-nut)”.

  1. Lines 102-103 - is there a difference between "yield" and "weight of green nuts"?

Reply: we have rewroten the sentence in new lines 96:

“Before examining the effects of morphological traits on green nut yield per tree and the Weight_100-nut”.

The green nut yield per tree and the Weight_100-nut reflected tree-level and nut-level harvesting traits, respectively.

  1. Line 133 - "mounts" should be changed to "amounts".

Reply: we change “mounts” to “amounts” in new line 130.

  1. Lines 149-150, 151 - are these correlations significant? This should be indicated.

Reply: they are significant. We have added statistical paramerter ( P value) to indicate the significance as in new lines 137-140:

“Spearman test indicated that green nut yield per tree was positively correlated with North-South crown width, crown area, and ground diameter (correlation coefficients were 0.4, 0.4, 0.3, respectively; P<0.05; Figure 2). In contrast, 100-nut weight was positively correlated with East-West crown width and ground diameter (both correlation coefficients were 0.2; P<0.05; Figure 2).”

  1. Figure 1 - is under crown height normally distributed or skewed?

Reply: it is skewed.

  1. Figure 2 - is there a significant negative correlation, i.e. is -0.2 significant and the light purple shade is to be interpreted as blue?

Reply: ‘-’ indicates a significant negative correlation. The color should be light purple. We have changed the description in new lines 290:

“The red and light purple colors in the grids indicate significant positive and negative correlations, respectively.”

  1. Figure 4 - the correlations are significant but very small in magnitude - are they of practical

Reply: Yes, the correlation is the only finding we found when we explored the traits and  Weight_100-nut, but it is indeed too small. We have moved the figure into the appendix.

Reviewer 2 Report

THE WORK MAY BE PUBLISHED IN THE JOURNAL ON THE SCREEN. YOU ONLY NEED TO COMPLY WITH THE SMALL CORRECTIONS AND SUGGESTIONS REFERRED TO THE ATTACHED MANUSCRIPT.

THE EDITOR WILL BE ABLE TO SOLVE IN WITHOUT PROBLEMS.

Author Response

Dear reviewer,

Thank you very much for the valuable comments and suggestions. We have revised the manuscript based on your suggestions as following. Moreover, the revised munuscript has been sent for an English check in MDPI.

  1. THIS ITEM ABSTRACT IS SEPARATED OF THE TEXT:

Abstract "As an important..."

Reply: we have rejusted the sentence in new lines 12-13:

“Abstract:

As an important”.

  1. line 87-88 WHY THIS UNIT hm?

1hm = 100 m

1hm2 = 100m x 100m = 10,000 m2 = 1 ha

10 hm2 = 10 x 10,000 = 100,000 m2 = 10 ha

Reply: We have changed “hm2” to “ha” in new lines 81-82.

  1. THE CONCLUSION MUST BE WRITTEN DIRECTLY AND WITHOUT EXPLAINING THE WHY. IF IT DOES'T STAY AS RESULTS AND A DISCUSSION. IN THIS ITEM 5, ONLY THESE LAST TWO ARE IN THE CORRECT FORMAT: "However, the two factors can only explain 23% of the yield variation, which indicates that the yield of Chinese Torreya may also be affected by other factors not monitored in this study. We (IT WAS IN REPLACEMENTT OF WE )speculate that it may be due to the differences in microclimate and soil fertility of Chinese Torreya growth, or the selected plant character indicators are not perfect, and the specific reasons need to be further studied. "

Reply: we have rewroten the conclusion new lines 181-189:

“Among all the investigated morphological traits, the ground diameter was the most significant positive influencing factor of the green nut yield of Chinese Torreya. Additionally, the ground diameter significantly and positively correlated with all the other traits except under_crown height, which may indicate that other morphological traits might indirectly affect the yield via the ground diameter. Through linear multiple regression analysis, we found that the two factors of North-South crown width and ground diameter entered the final model of green nut yield prediction of Chinese Torreya (P < 0.01). However, the two factors can only explain 23% of the green nut yield variation, which indicates that the green nut yield of Chinese Torreya may also be affected by other factors not monitored in this study. We speculate that it may be due to the differences in microclimate and soil fertility of Chinese Torreya individuals, which need to be further studied in the future.”

Reviewer 3 Report

The paper has looked at a very important area that can be used in the improvement of the Chinese Torreya. I enjoyed reviewing the paper, however, there is need to improve in the following areas:

(1) there is need to briefly describe the experimental design in the abstract

(2) the conclusion can be improved by summarizing the messages from the observed results.

(3) some suggestions are highlighted in the attached reviewed manuscript

Author Response

Dear reviewer,

Thank you very much for your comments and suggestions. We have revised the manuscript as following. Moreover, the revised manuscript has been sent for an English check in MDPI.

  1. “RMB” write in full for the first time

Reply: we change “RMB” to “of Renminbi” in new line 38.

  1. “hm2” verfy these units???

Reply: we change “hm2” to “ha” in new line 81-82.

  1. “data” upcase “d”

Reply: we change “data” to “Data” in new line 98.

  1. which ones?

Reply: we have rewroten the sentence in new lines 98-99:

“the following statistical examinations (Spearman test on corelations between traits, and linear regressions between green nut yield and morphological traits)”

  1. “individual” upcase “i”

Reply: we change “individual” to “Individual” in new line 122.

  1. “Height” upcase “h”

Reply: we change “Height” to “height” in new line 139.

  1. “don’t” to “do not”

Reply: we change “don’t” to “do not” in new line 177.

  1. why are you suggesting this, support with your results? where is this model??? The model should be shown first and then this statement later

Reply: we have deleted this part “In the breeding of Chinese Torreya, the ground diameter,  crown area and green nut yield should be considered comprehensively, and the ground diameter should be considered emphatically. The structural equation model constructed in this study is a linear model, which describes the linear relationship between green nut yield and morphological traits of Chinese Torreya. There may also be a nonlinear relationship among these variables, so the nonlinear relationship among the above variables needs further study.”.

  1. in line 205-219 These are more of results rather than conclusion. What is the message in these findings?

Reply: we have rewroten the conclusion new lines 181-189:

“Among all the investigated morphological traits, the ground diameter was the most significant positive influencing factor of the green nut yield of Chinese Torreya. Additionally, the ground diameter significantly and positively correlated with all the other traits except under_crown height, which may indicate that other morphological traits might indirectly affect the yield via the ground diameter. Through linear multiple regression analysis, we found that the two factors of North-South crown width and ground diameter entered the final model of green nut yield prediction of Chinese Torreya (P < 0.01). However, the two factors can only explain 23% of the green nut yield variation, which indicates that the green nut yield of Chinese Torreya may also be affected by other factors not monitored in this study. We speculate that it may be due to the differences in microclimate and soil fertility of Chinese Torreya individuals, which need to be further studied in the future.”
